# Prevalence and determinants of care needs among older people in Ghana

**Kofi Awuviry-Newton[1]\*, Kwadwo Ofori-Dua[2], Charles Selorm Deku[2], Kwamina Abekah-Carter[3], Victoria Awortwe[4], George Ofosu Oti[3]**

**1** African Health and Ageing Research Centre, Winneba, Central Region, Ghana, **2** Department of Sociology and Social Work, Kwame Nkrumah University of Science and Technology, Kumasi, Ghana, **3** Department of Social Work, University of Ghana, Accra, Ghana, **4** Department of Social Studies, Universitetet I Stavanger, Stavanger, Norway

\* newscous@gmail.com

## Abstract

### Introduction

Given the longevity noticed among older people in Ghana, and the potential occurrence of functional disability in later years of life, it has become essential to understand their care needs. This study examined the care needs in daily tasks and associated factors in Ghana, following the World Health Organisation International Classification of Functioning, Disability and Health framework.

### Materials and methods

A cross-sectional survey was conducted among a sample of 400 older people from Komfo Anokye Teaching Hospital in Southern Ghana. Care need was assessed by one question; "Do you regularly need help with daily tasks because of long-term illness, disability, or frailty?" Multivariate logistic regression was used to test the association between care need and independent variables based on the WHO-ICF conceptual framework.

### Results

Majority of the sample (81%), particularly women (54%) reported needing care in daily tasks. Per the WHO-ICF conceptual framework, functional disability—activity variable, (OR = 1.07 95%CI: 1.05–1.09, p<0.001), and absence of government support—an environmental factor, (OR = 3.96 95%CI: 1.90–8.25, p<0.001) were associated with care need.

### Conclusions

The high prevalence of care needs among older people may offer an indication that majority of older people in Ghana could benefit from long-term care services. Functional disability and the absence of government support are the major issues that need to be prioritised in addressing the increased demand for care related to performing daily tasks among older people in Ghana.

**Data Availability Statement:** All relevant data are within the manuscript (Appendices) and its Supporting Information files.

**Funding:** No - Include this sentence at the end of your statement: The funders had no role in study design, data collection and analysis, decision to publish, or preparation of the manuscript.

**Competing interests:** The authors have declared that no competing interests exist.

## Introduction

The population of people aged 60 years or older in Ghana is growing rapidly due to decreasing birth rates and delayed mortality [1]. The number of older people in Ghana increased more than seven-fold from 213,477 (4.5%) in 1960 to 1,643,381 (6.7%) in 2010 [2]. The percentage of older people in Ghana is further expected to increase to 9.8% by 2050 [2]. Although the percentage of older people in Ghana is lower than that of some developed countries, it is worthy to note that it is growing at an alarming rate [3]. Population ageing has an association with the increase in service need and thus, requires updated policies and programs to respond to the current and future health needs of older people in Ghana [4], particularly those who may be living with functional disabilities- i.e. limitation in performing life activities such as bathing.

Given the increase in longevity among older people in Ghana [1, 2], and the likely occurrence of functional disability in older people in later years [5], it is relevant to explore their care needs. Care needs can be identified as the need for assistance in essential life activities, including activities of daily living (ADL) and instrumental activities of daily living (IADL) [6, 7]. Independence in ADLs (defined as basic self-care tasks, such as bathing) and IADLs (secondary task to care for self, as well as home responsibilities) are essential to promote the health and social wellbeing of older people. What is unknown in Ghana is the prevalence of care needs and its associated factors among older people. Understanding the care needs related to daily living tasks among older people in Ghana is essential to provide data to assist policy and program developers to provide the appropriate health and social interventions. In Ghana, the declining traditional extended family support system for older people [8, 9] has heightened the need to understand the prevalence and associated factors of older people's care needs [8, 10]. Nonetheless, adult children and other family relatives still assume all the roles as primary caregivers for older people who may need care [9, 11] since formal social care is yet to gain prominence in the Ghanaian context. Often, these caregivers offer care and support with high cost and burden [12], casting doubt on the continuity of their care.

Available international evidence on factors associated with high care needs are reported as personal, health and environmental factors [13–15]. For instance, Rosso, Auchincloss [16] study reported that transportation and mobility difficulties are associated with high care needs among older people. Similarly, other studies identified disability-unfriendly physical environment as a factor accounting for high care needs among older people [13, 14]. Personal factors, such as living in an urban area [17], being a woman [18], living closer to one's children [19–21], advanced age and living alone marital status [15], and being divorced [22] are associated with high care needs. Furthermore, low level of education is associated with high care needs of older people [23]. Health-related factors associated with high care needs among older people are reporting health as bad and living with depression [13, 19, 24], as well as living with Alzheimer's disease or related dementia [25]. However, studies on determinants of care needs of older people in Africa is scanty [26, 27], with none in Ghana. In this study, care needs will be considered as a body function and structural impairment, which may be determined by intrinsic and extrinsic factors surrounding older people. This current study contributes knowledge to supplement existing literature by exploring care needs in the context of the World Health Organisation International Classification of Functioning (WHO-ICF) components. This information will enhance our understanding of the holistic factors determining older people's functioning level. We hypothesized that the higher the needs across the components of the WHO-ICF, the high care older people would need.

### Analytical framework

According to the World Health Organisation, the WHO-ICF is a conceptual framework that is essential for describing and understanding a person's health in terms of function and disability

[28]. Function, in this framework, refers to all body functions, activities and participation without restrictions [28]. Disability refers to impairments, activity limitations, and participation restrictions [28]. The WHO-ICF acknowledges the interaction between determinants of health and disability, as well as personal and environmental factors [28, 29]. This framework perceives that the interactions of these entire components help in understanding the health needs of people, including older people. Accordingly, it is hypothesised in this study that factors determining the care needs of older people will encompass environmental, health status, community, and network factors, as well as their intrinsic capacities of older people. Using the WHO-ICF in examining care needs among older people, will reveal the various factors potentially influencing the current and future care needs of older people in Ghana.

## Methods

### Study design, setting and sample

A cross-sectional survey was conducted at Komfo Anokye Teaching hospital located in the southern part of Ghana. The study required a sample of 200 at a confidence level of 95% using Epi Info software (version 7.2.3). However, we increased this to 400 to compensate for any probable loss of response for questions included in this study. The participants were older people aged 60 years or over, admitted to the hospital due to any health problem or frailty, and who stayed a minimum of one night in any ward of the hospital, and who gave their permission to participate in the study. Older people who were seriously sick were excluded from the study. The study employed a consecutive sampling technique based on the hospital admission register to recruit participants.

Recruitment of participants was completed on randomly selected days (i.e., 4 days per week) over eight hours to increase the likelihood of older people who were admitted being offered the opportunity to participate. After nurses had determined the eligibility of older people, the primary researcher sought their consent and collected the survey data. Data were collected during participants' hospitalisation in the time and day chosen by the participants.

### Data collection

Data were collected between the months of April and August 2018. A survey questionnaire was used to collect data from all participants by the primary researcher (see Appendix A in S1 Appendix). A self-administered questionnaire was used to collect data from literate participants. The primary researcher read the questions to solicit responses from participants who could not read due to their low educational level. The questionnaire consists of the socio-demographic profile, information related to their hospital admission, care needs and functional ability level using the WHO global Study on AGEing and adult health questionnaire.

### Measures

#### Dependent variable

**Care needs.** Care needs was assessed by one question; "Do you regularly need help with daily tasks because of long-term illness, disability or frailty (e.g., personal care, getting around, preparing meals, etc.?)" based on dichotomous responses "Yes" and "No".

**Independent variables.** Based on the WHO-ICF, the six domains with their respective variables are discussed below:

**Personal factors.** According to the WHO-ICF, personal factors refers to the intrinsic nature of individuals [29]. Personal factors included age, marital status, education, religion, and employment status. In this study, it was hypothesised that advanced age, being a female,

living as a widow, having no religious affiliation, having a low level of education, living in an urban area, living alone, and not working were associated with high care needs.

Age was measured as a continuous variable. Gender was measured on a categorical nominal variable "male" or "female". Marital status was measured with "never married", "married/ cohabiting", "separated/divorced", and "widowed". Marital status was categorised as "single", "separated or divorced", "married or cohabiting", and "widowed" for chi-square analysis. In terms of education, the response categories were "no education", "at most Junior High completed", "Senior High completed" and "at least College/Pre-University completed". This was further categorised as "no education", "at most Junior High completed", and "at Least Senior High School completed". Religion was initially measured as "none", "Christianity (including Roman-Catholics)", "Islam", and "Traditional religion". However, for the analysis, the "none" and "Traditional religion" were combined as "none", with "Christianity" and "Islam" treated as separate categories. In terms of living arrangements, the categories were "alone", "as a couple", "as a couple and with children". Employment status was categorised as "currently working" and "currently not working".

**Body function and structure.**   Body function and structure refers to the level of impairment or function of individuals that can influence their overall health [29]. The body function and structure was assessed by one open-ended question "What physical impairment did the doctor/nurse diagnose you with . . . . . . . . ..?", was used to determine any impairment. The responses reflecting this component was mainly "visual impairment" and "injury". These variables were coded as "1" meaning "Yes" and "0" meaning "No".

**Health condition.**   Health condition refers to any illness or chronic condition that can affect the overall functioning of older people [29]. In this study, health conditions were assessed by one open-ended question "What illness did the doctor/nurse diagnose you with. . .?" These conditions include chronic conditions, infectious diseases and alcoholism. The health conditions mainly reported by participants were diabetes, stroke, ulcers, cancer, hypertension, kidney disease, asthma, heart disease, and lung diseases. Other health conditions reported were hernia, malaria, chronic alcoholism, heart failure, jaundice, atrial fibrillation, ganglia, urosepsis, haemorrhage, gastroenteritis, uremic encephalopathy, anaemia, urinary tract infection, cardiomyopathy, goitre, chronic left external capsular infarct, appendicitis, liver disease, pneumonia, intestinal lung disease, cellulitis, gangrene, hepatitis, cataract, fall injury, kidney disease, blindness, fibroid, angina, prostate enlargement, and neurological problem. Each of these variables was categorized as "1" meaning "Yes" if the condition was present and "0" meaning "No" if the condition was not present. For the analysis, these conditions were transformed into multi-morbidity variables coded as "no condition", "only one condition" and "two or more conditions".

**Activity limitations.**   *Functional disability*. According to the WHO-ICF, functional disability is an activity limitation which refers to difficulties experienced by people in carrying out life activities [29]. Functional disability was scored by a 24 items questionnaire, consisting of ADL, and IADL. Difficulties were assessed using ordinal response categories (none (0), mild (1), moderate (2), severe (3), extreme (4)) in response to the question "In the last 30 days, how much difficulty did you have with . . . . . . . .due to health problems, injuries, mental or emotional problems?" Internal consistency of response across the 24 items in this study was assessed and reliability was found to be high (Cronbach alpha = 0.96).

**Environmental factors.**   According to the WHO-ICF, environmental factors refer to the physical, social and attitudinal aspects that influence individual function [29]. In this study, the social environment was assessed by two questions, *(1) "How many children do you have*? The response was categorised as "none", "1–5", and "6 or more", and *(2)" Do you receive support from neighbours or community*? *Do you receive support from the government*? *Do you*

*receive support from religious members*? *Do you receive support from religious members*? *Do you receive support from non-governmental organisations*? Each response was categorised as "yes" if the support was present and "no" if no support from that source. Emotional support was assessed by one question "*How many times during the past week did you spend time with someone who does not live with you, that is, you went to see them, or they came to visit you, or you went out together*?" based on eight points Likert Scale (0 to 7) (e.g. Frequency was assessed as 0 to 7 or more, but was categorised as "none", "1–5 times" and "6 or more times" for analysis. Perceived social support was measured using nine questions based on three-point scales "hardly ever", "some of the time", and "most of the time". For example, *"Does it seem that your family and friends, people who are important to you understand you*?

### Participation

Participation refers to the level of engagement in social or community activities to increase functioning [29]. In this study, the participation in social or community activities was assessed by one question: *"About how often did you go to meetings of clubs, religious meetings or other groups that you belong to in the past week*?", based on a scale from "none" to "seven or more". These responses were categorized as "none", and "at least once".

### Data analysis

Descriptive statistics were used to describe the demographic information of study participants. To compare relationships between categorical variables, a bivariate analysis was used. Bivariate and multivariable logistic regressions were performed to assess any significant relationship between variables under WHO-ICF components (independent variables) and care needs (dependent variable). Logistic regression was used to estimate crude and adjusted odds ratios and 95% confidence intervals to test for associations between the dependent and independent variables. Any variable with a p-value of 0.2 in the bivariate association was considered to include in the multivariate logistic regression analysis. A P-value of 0.05 was used to identify the determinants of care needs. Stata version 15 was used to manage the analysis.

### Ethical consideration

We received ethics approval from the Kwame Nkrumah University of Science and Technology Ethics Committee (CHRPE/RC/033/18) before the commencement of this study per the Declaration of Helsinki. Anonymity and confidentiality of the study site and participants were also ensured.

## Results

The demographic characteristics of the participants are presented in Table 1. Participants were mainly women (51%), and the average age was 72 years. Furthermore, 81% of participants reported a need for care in daily living tasks (See Table 1).

### Bivariate analysis of care needs with variables across WHO-ICF components

We included variables that had p<0.2 in the multivariate logistics regression for further analysis. From the bivariate analysis, marital status, religious belief, residence, living arrangement, visual impairment, injury, perceived support variables, religious group/member support, non-governmental organisation support, and several children were not significant with care needs. More than half (51%) of the participants who reported needing care lacked governmental support (see Table 2).

**Table 1. Demographic characteristics of older people, and care needs.**

| Demographic Characteristics (N = 400) | N (%) |
|---|---|
| Age (mean, SD) | 71.3±8.42 |
| **Sex** | |
| Male | 196 (49.0 |
| Female | 204 (51.0) |
| **Marital status** | |
| Single/separated/divorced | 58 (14.5) |
| Currently married/cohabiting | 212 (53.0) |
| Widowed | 130 (32.5) |
| **Education** | |
| No education | 128 (32.0) |
| At maximum junior high completed | 209 (52.3) |
| At least senior high completed | 63 (15.8) |
| **Religion** | |
| None | 27 (6.75) |
| **Christianity** | 331 (82.8) |
| Islam | 42 (10.0) |
| **Residence** | |
| Rural | 227 (56.8) |
| Urban | 173 (43.3) |
| **Living arrangement** | |
| Alone | 50 (12.5) |
| With couple | 100 (25.0) |
| With couple and children | 250 (62.5) |
| **Employment status** | |
| Currently working | 156 (39.0) |
| Currently not working | 244 (61.0) |
| **Do you need care?** | |
| Yes | 322 (80.5) |
| No | 78 (19.5) |

## Determinants of care needs among older people based on the WHO-ICF

In the unadjusted logistic regression, age, gender, education, employment status, multi-morbidity, disability score, participation restriction status, speaking with someone via the phone, spending time with someone whom older people do not live with, and government support were statistically significantly associated with older people' need for care (see Table 3).

However, adjusting for all these statistically significant variables, disability score and government support were independently statistically significantly associated with the care needs of older people.

Regarding the disability score, a 1-unit increase in disability score increases the older people's need for care by 7%. Older people who lack government support was 3.96% more likely to report a need for care.

## Discussion

To the best of our knowledge, this is the first study to select variables based on the WHO-ICF framework to study care needs among older people in Ghana. Overall, the findings revealed that care needs are related to environmental and activity limitation components of the

**Table 2. Bivariate analysis of care needs across WHO-ICF components.**

| Demographic Characteristics (N = 400) | Total N (%) | care need | | p-value |
|---|---|---|---|---|
| PERSONAL FACTORS | | Yes N (%) | No N (%) | |
| Age *(mean, SD)* | 71.3±8.42 | 71.8±8.45 | 68.9±7.93 | 0.006 |
| Sex | | | | 0.027 |
| Male | 196 (49.0) | 149 (46.3) | 47 (60.3) | |
| Female | 204 (51.0) | 173 (53.7) | 31 (39.7) | |
| Marital status | | | | 0.623 |
| Single/separated/divorced | 58 (14.5) | 45 (14.0) | 13 (16.7) | |
| Currently married/cohabiting | 212 (53.0) | 169 (52.5) | 43 (55.1) | |
| Widowed | 130 (32.5) | 108 (33.5) | 22 (28.2) | |
| Education | | | | 0.006 |
| No education | 128 (32.0) | 110 (34.2) | 18 (23.1) | |
| At least junior high completed | 209 (52.3) | 170 (52.8) | 39 (50.0) | |
| At least senior high completed | 63 (15.8) | 42 (13.0) | 21 (26.9) | |
| Religion | | | | 0.933 |
| None | 27 (6.75) | 21 (6.52) | 6 (7.69) | |
| **Christianity** | 331 (82.7) | 267 (82.9) | 64 (82.1) | |
| Islam | 42 (10.5) | 34 (10.6) | 8 (10.3) | |
| Residence | | | | 0.659 |
| Rural | 227 (56.8) | 181 (56.2) | 46 (59.0) | |
| Urban | 173 (43.3) | 141 (43.8) | 32 (41.0) | |
| Living arrangement | | | | 0.842 |
| Alone | 50 (12.5) | 41 (12.7) | 9 (11.5) | |
| With couple | 100 (25.0) | 82 (25.5) | 18 (23.1) | |
| With couple and children | 250 (62.5) | 199 (61.8) | 51 (65.4) | |
| Employment status | | | | 0.006 |
| Currently working | 156 (39.0) | 115 (35.7) | 41 (52.6) | |
| Currently not working | 244 (61.0) | 207 (64.3) | 37 (47.4) | |
| BODY FUNCTION AND STRUCTURE | | | | |
| Visual impairment | | | | 0.268 |
| Yes | 5 (1.25) | 5 (1.55) | 0 (0.00) | |
| No | 395 (98.8) | 317 (98.5) | 78 (100) | |
| Injury | | | | 0.212 |
| Yes | 265 (66.3) | 218 (67.7) | 47 (60.3) | |
| No | 135 (33.8) | 104 (32.3) | 31 (39.7) | |
| CHRONIC HEALTH CONDITION | | | | |
| Multi-morbidity | | | | 0.138 |
| At most 1 condition | 308 (77.0) | 243 (75.5) | 65 (83.3) | |
| Any 2 or more conditions | 92 (23.0) | 79 (24.5) | 13 (16.7) | |
| ACTIVITY LIMITATION | | | | |
| Disability score (mean, SD) | 54.6±21.2 | 60.1±16.9 | 31.9±21.9 | <0.001 |
| ENVIRONMENTAL FACTORS | | | | |
| *Perceived Support* | | | | |
| Family and friend understand you | | | | 0.809 |
| Hardly ever | 74 (19.0) | 62 (19.6) | 12 (16.4) | |
| Some of the time | 162 (41.7) | 130 (41.1) | 32 (43.8) | |
| Most of the time | 153 (39.3) | 124 (39.2) | 29 (39.7) | |
| Feel useful to family and friends | | | | 0.742 |

*(Continued)*

**Table 2.** (Continued)

| Demographic Characteristics (N = 400) | Total N (%) | care need | | p-value |
|---|---|---|---|---|
| Hardly ever | 83 (20.8) | 68 (21.1) | 15 (19.2) | |
| Some of the time | 173 (43.3) | 141 (43.8) | 32 (41.0) | |
| Most of the time | 144 (36.0) | 113 (35.1) | 31 (39.7) | |
| **Awareness of matters concerning family and friends** | | | | 0.381 |
| Hardly ever | 179 (44.8) | 142 (44.1) | 37 (47.4) | |
| Some of the time | 112 (28.0) | 95 (29.5) | 17 (21.8) | |
| Most of the time | 109 (27.3) | 85 (26.4) | 24 (30.8) | |
| **Share deepest problems with some family and friends** | | | | 0.387 |
| Hardly ever | 83 (20.8) | 67 (20.8) | 16 (20.5) | |
| Some of the time | 173 (43) | 144 (44.7) | 29 (37.2) | |
| Most of the time | 144 (36.0) | 11 (34.5) | 33 (42.1) | |
| *Emotional support* | | | | |
| **Often time you spoke with someone via telephone (past week)** | | | | <0.001 |
| None | 173 (43.3) | 154 (47.8) | 19 (24.4) | |
| 1–5 times | 161 (40.3) | 123 (38.2) | 38 (48.7) | |
| 6 or more times | 66 (16.5) | 45 (14.0) | 21 (26.9) | |
| **Spent time with someone who does not live with you (past week)** | | | | 0.034 |
| None | 54 (13.5) | 47 (14.6) | 7 (8.97) | |
| 1–5 times | 196 (49.0) | 164 (50.9) | 32 (41.0) | |
| 6 or more times | 150 (37.5) | 111 (34.5) | 39 (50.0) | |
| **Neighbours/community support** | | | | 0.110 |
| Yes | 222 (55.5) | 185 (57.5) | 37 (47.4) | |
| No | 178 (44.5) | 137 (42.6) | 41 (52.6) | |
| **Government support** | | | | <0.001 |
| Yes | 174 (43.5) | 158 (49.1) | 16 (20.5) | |
| No | 226 (56.5) | 164 (50.9) | 62 (79.5) | |
| **Religious group/members support** | | | | 0.216 |
| Yes | 235 (58.8) | 194 (60.3) | 41 (52.6) | |
| No | 165 (41.3) | 128 (39.8) | 37 (47.4) | |
| **Non-government organisation support** | | | | 0.443 |
| Yes | 10 (2.50) | 9 (2.80) | 1 (1.28) | |
| No | 390 (97.5) | 313 (97.2) | 77 (98.7) | |
| **Number of children** | | | | 0.554 |
| At most one child | 40 (10.0) | 31 (9.63) | 9 (11.5) | |
| 2–4 | 128 (32.0) | 100 (31.1) | 28 (35.9) | |
| 5 or more | 232 (58.0) | 191 (59.3) | 41 (52.6) | |
| **PARTICIPATION** | | | | |
| **Often times you attend meetings (past week)** | | | | 0.006 |
| None | 313 (78.3) | 261 (81.1) | 52 (66.7) | |
| At least once | 87 (21.8) | 61 (18.9) | 6 (33.3) | |

WHO-ICF framework. The study revealed a high prevalence of care needs among older people, particularly in women. Functional disability and absence of government support to older people were associated with high care needs among older people.

The prevalence of care needs was high among older people, and this implies that the majority of older people in Ghana may need care, especially assistance in fulfilling daily activities. This finding echoes the reasons why most older people report needing caregivers [30] and it

**Table 3. Determinants of care needs among older people in Ghana.**

| Variables based on ICF component | Unadjusted OR; 95% CI | Adjusted OR, 95% CI |
|---|---|---|
| **PERSONAL FACTORS** | | |
| **Age** | 1.05 (1.01, 1.08)** | 1.01 (0.97, 1.06) |
| **Gender** | | |
| Female (vs male) | 1.76 (1.06, 2.91) * | 0.99 (0.48, 2.10) |
| **Education** | | |
| No education (vs At least senior high completed) | 3.06 (1.48, 6.30)** | 1.83 (0.66, 5.04) |
| at most junior high completed (vs At least senior high completed) | 2.18 (1.16, 4.09)* | 1.43 (0.60, 3.37) |
| **Employment status** | | |
| Currently not working (vs Currently working)) | 2.00 (1.21, 3.29)** | 1.61 (0.80, 3.22) |
| **HEALTH CONDITION** | | |
| Multi-morbidity | | |
| Any 2 or more conditions (vs at most one condition) | 1.63 (0.85, 3.11) | |
| **ACTIVITY LIMITATION** | | |
| Disability score | 1.07 (1.05, 1.08)*** | 1.07 (1.05, 1.09)*** |
| **PARTICIPATION RESTRICTION** | | |
| Often times you attend meetings (past week) | | |
| None (vs at least once) | 2.14 (1.24, 3.70)** | 0.76 (0.32, 1.79) |
| **ENVIRONMENTAL FACTORS** | | |
| *Emotional support* | | |
| **Spent time with someone who does not live with you (past week)** | | |
| None (vs6 or more times) | 2.36 (0.99, 5.65) | 1.84 (0.62, 5.44) |
| 1–5 times (vs 6 or more times) | 1.80 (1.06, 3.05)* | 1.68 (0.83, 3.40) |
| **Often time you spoke with someone via telephone (past week)** | | |
| None (vs 6 or more times) | 3.78 (1.87, 7.65)*** | 1.78 (0.70, 4.50) |
| 1–5 times (vs 6 or more times) | 1.51 (0.80, 2.85) | 2.33 (0.99, 5.46) |
| **Neighbours/community support** | | |
| No (vs Yes) | 0.67 (0.41, 1.10) | |
| **Government support** | | |
| No (vs Yes) | 3.73 (2.07, 6.74)*** | 3.96 (1.90, 8.25)*** |

Significant at

*p-value < 0.05

**p-value < 0.01

***p-value<0.001.

justifies why they often express concerns about caregiver availability in developing countries [31]. This finding is a stepping-stone to inform policymakers, social welfare program developers and health care professionals in Ghana to develop innovative ideas or approaches to meet the long-term care needs of older people. These interventions should take into consideration the gender variation in care needs as older women required more care compared to men [32]. Population ageing in Ghana is a new phenomenon, and so knowing the prevalence of care needs is essential to developing programs and services to meet the care needs of older people.

The findings that one additional increase in disability score is associated to a 7% increase in care needs reveal how health-related factors can decrease the quality of life of older people in Ghana. This finding is consistent with previous studies conducted globally [13, 14]. Older people living with a functional disability might experience restrictions in participating in everyday activities they may cherish [33]. Activities needed to improve older people' health, such as

visiting friends and attending social gatherings may be impacted [33], thus, increasing the need for care. Reliable health insurance programs should be established by mandated state institutions to increase the accessibility of healthcare among older people in Ghana. Additionally, there should be a promotion of national interest in the health needs of older people to enhance older people's independence and wellbeing.

In this study, it was further revealed that government support is significantly associated with older people's need for care. By implication, the absence of government support of any kind, be it health, financial, emotional, or physical will mean older people will have a high need for care. This is understandable in Ghana because the traditional extended family that was providing care and support for older people in times of disability is gradually depleting [8, 9], drawing attention to the need for state to provide support for older people. The absence of government support or care may be catastrophic for older people in the future because they will be left alone to care for themselves. This finding demonstrates how health and social welfare programs should be strengthened to attend to the diverse needs of older people in Ghana.

In this study, personal factors, such as advanced age, lower level of education being divorced or living alone were not statistically significant in the adjusted model. This finding is contrary to other studies that revealed a significant relationship between these variables and higher care needs in older people [15, 22, 23]. The difference in the relationship between age and care needs may be because participants were recruited from a hospital and were all aged 60 years or older. So, their care needs may not be significantly influenced by their age but rather their functional difficulty.

The current study connotes several implications for future research. First, the study's findings present evidence for researchers to explore the government's interest in addressing the long-term care needs of older people through the provision of health and social services. It also implies that more longitudinal research needs to be conducted on functional disabilities among older people in Ghana. Qualitative studies exploring the narratives of older people concerning their functional disabilities and need for care are essential, and offer stakeholders the urgent need to intervene in the welfare of older people in Ghana.

The study has some strengths and limitations that need to be acknowledged. In terms of strength, this is the first study to explore the prevalence of care needs and associated factors using a conceptual framework such as WHO-ICF. The limitation is that the participants were a small, hospitalised sample, and the findings may not apply to the general population of older people in Ghana. Moreover, this study did not model the interaction across the domains of the WHO-ICF. However, this was necessary because the aim was to use the WHO-ICF as a conceptual guide for variable selections. This study did not include the physical environment that may also determine the level of care needs in older people. More research to model the various components of the WHO-ICF is essential to understand the overall impact on care needs.

## Conclusion

A high prevalence of care needs exists among older people particularly women in Ghana. Employing the WHO-ICF, enabled us to identify the environmental and health-related predictors associated with care needs among older people in Ghana. Interventions to improve the functional abilities of older people and increase the national interest in the care needs of older people is needed in Ghana. These findings have drawn attention to the multi-sectorial interest in the health and social care needs of older people in Ghana. Therefore, mandated institutions should make a conscious effort to make available formal social care programmes to assist older people to meet their care needs.

## Supporting information

**S1 Appendix.**
(DOCX)

**S1 Dataset.**
(DTA)

## Author Contributions

**Conceptualization:** Kofi Awuviry-Newton, Kwadwo Ofori-Dua, Charles Selorm Deku, Kwamina Abekah-Carter, Victoria Awortwe, George Ofosu Oti.

**Data curation:** Kofi Awuviry-Newton.

**Formal analysis:** Kofi Awuviry-Newton.

**Investigation:** Kofi Awuviry-Newton.

**Methodology:** Kwadwo Ofori-Dua, Charles Selorm Deku, Victoria Awortwe.

**Project administration:** Kofi Awuviry-Newton.

**Software:** Kofi Awuviry-Newton.

**Supervision:** Kwadwo Ofori-Dua.

**Validation:** Kwadwo Ofori-Dua, Charles Selorm Deku.

**Writing – original draft:** Kofi Awuviry-Newton.

**Writing – review & editing:** Kwadwo Ofori-Dua, Charles Selorm Deku, Kwamina Abekah-Carter, Victoria Awortwe, George Ofosu Oti.

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
