## [Decision Letter · Decision Letter 0]

3 Sep 2021

PONE-D-21-10419

Prevalence and determinants of care needs among older people in Ghana

PLOS ONE

Dear Dr. Abekah-Carter,

Thank you for submitting your manuscript to PLOS ONE. After careful consideration, we feel that it has merit but does not fully meet PLOS ONE’s publication criteria as it currently stands. Therefore, we invite you to submit a revised version of the manuscript that addresses the points raised during the review process.

We look forward to receiving your revised manuscript.

Kind regards,

Mohammad Bellal Hossain

Academic Editor

PLOS ONE

Journal Requirements:

6. We noticed you have some minor occurrence of overlapping text with the following previous publication, which needs to be addressed:

- https://journals.plos.org/plosone/article?id=10.1371%2Fjournal.pone.0233541

The text that needs to be addressed involves the  "Health Conditions" and "Personal Factors" parts of the Methods section. 

In your revision ensure you cite all your sources (including your own works), and quote or rephrase any duplicated text outside the methods section. Further consideration is dependent on these concerns being addressed.

Reviewers' comments:

Reviewer's Responses to Questions

**Comments to the Author**

1. Is the manuscript technically sound, and do the data support the conclusions?

Reviewer #1: Partly

Reviewer #2: Yes

2. Has the statistical analysis been performed appropriately and rigorously? 

Reviewer #1: I Don't Know

Reviewer #2: Yes

3. Have the authors made all data underlying the findings in their manuscript fully available?

Reviewer #1: No

Reviewer #2: Yes

4. Is the manuscript presented in an intelligible fashion and written in standard English?

Reviewer #1: No

Reviewer #2: Yes

5. Review Comments to the Author

Reviewer #1: Review of plos one manuscript

introduction

1. What is the difference between instrumental ADL and ADL. Readers need to understand these terms in the introduction.

2. There is a lot mentioned about functional disability among the aged in introduction. If this is the main factor explored, then it must be mentioned in title and abstract in like manner. See highlighted portions. I see authors used a framework WHO-ICF in conducting the study. If there were specific factors such as functional disability, ADL etc in this framework that were explored, this should be made clearer in the introduction. Probably this work was about exploring those factors within the WHO-ICF framework.

3. In paragraph 3, authors stated that there is paucity of knowledge about……..this should come at the end of introduction, and then this can set the tone for the objectives of this study.

4. In paragraph 4, the authors bring in history of how Ghanaians lived in the past. This does seem to fit in here. I believe the authors want to make the point about the breakdown of family support systems for older Ghanaians in contemporary Ghana. This point can be made in a concise manner.

5. Again I do not see the point about care givers reporting poor health and psychological instability etc………..in paragraph 4. May be authors should concentrate on older people themselves and what the evidence is, about their challenges.

6. In paragraph 5, the authors state ‘globally, there are a number of studies…..’ the introduction has to be rearranged. This global piece was brought in after issues in Ghana were earlier introduced. Typically, the global aspects will be introduced then narrowed down to Africa and Ghana. This will make reading of manuscript better.

7. Again authors bring in WHO-ICF framework closer to the end of introduction. This does not quite fit in here. This should be bought up earlier and discussed in a way that shows the reasons for its usage.

Methods

1. The authors state that the minimum sample size required in this study was 200. Readers are unable to tell how this was arrived at. Which sample calculation method was used? The sampling techniques used are unclear. In one aspect, authors state consecutive sampling using hospital register and later states random selection of patients based on days they visited the hospital. This part should be clarified in methods.

2. Data collection. It will help if authors can add questionnaire as an appendix.

Plagiarism check.

This work has a 52% similarity with other works (checked with ithenticate). Authors should kindly modify to reduce the percentage of similarity with other published works.

Ethical considerations

I am unable to find statement on ethics in this work.

Reviewer #2: i. In the abstract, kindly indicate the research gap calling for the study.

ii. Under the data analysis section, the authors indicate that “ any variable with a p-value of 0.2 in the bi-variate association was considered for inclusion in the multivariate analysis” Is there any basis or theoretical justification for this? This should be discussed in the manuscript.

iii. In the methods, I did not see how the authors did measure validity and reliability of the data collection tool/ tool

iv. In the discussion, this is how the authors should discuss the results on the prevalence of care needs.

a. The authors should tell readers whether the prevalence of care needs reported in this study is higher than what has been reported elsewhere and then assign reasons for the differences. The authors should tell readers whether the prevalence of care needs reported in this study is the same as what has been reported elsewhere.

b. The authors should tell readers whether the prevalence of care needs reported in this study is lower than what has been reported elsewhere and then assign reasons for the differences.

v. A paragraph of the discussion should be devoted for the specific contribution of the study to literature.

vi. The authors should also discussed the strengths of the study.

vii. The authors should further discuss how the limitations of the study did not affect the findings of the study.

viii. In the conclusion section, the authors did indicate that the prevalence of care needs is high among hospitalized older people. Is there any theoretical or conceptual basis for determining the prevalence at which care needs are high?

6. PLOS authors have the option to publish the peer review history of their article (what does this mean?). If published, this will include your full peer review and any attached files.

Reviewer #1: No

Reviewer #2: No

---

## [Author Response · Author response to Decision Letter 0]

29 Oct 2021

Reviewer #1: 

1. What is the difference between instrumental ADL and ADL? Readers need to understand these terms in the introduction.

Response

Authors have clarified the differences between ADL and IADLs. The sentence now reads as: “Independence in ADLs (defined as basic self-care, tasks such as bathing) and IADLs (secondary task to care of self and home household responsibilities) are essential to promote the health and social wellbeing of older people.”

2. There is a lot mentioned about functional disability among the aged in introduction. If this is the main factor explored, then it must be mentioned in title and abstract in like manner. See highlighted portions. I see authors used a framework WHO-ICF in conducting the study. If there were specific factors such as functional disability, ADL etc in this framework that were explored, this should be made clearer in the introduction. Probably this work was about exploring those factors within the WHO-ICF framework.

In this study, the authors alluded to functional disability as helping to understand the care needs of older adults and supported this with relevant literature from the western countries. It was then studied as an independent variable that was potentially associating with care needs. The WHO-ICF framework was used in the categorisation of the independent variables. The According to the framework, the functional disability variable falls under the “activity limitation” domain. 

3. In paragraph 3, authors stated that there is paucity of knowledge about……..this should come at the end of introduction, and then this can set the tone for the objectives of this study.

This section has been removed and sent to the conclusion of the introduction. 

4. In paragraph 4, the authors bring in history of how Ghanaians lived in the past. This does seem to fit in here. I believe the authors want to make the point about the breakdown of family support systems for older Ghanaians in contemporary Ghana. This point can be made in a concise manner.

Thanks for this comment. Authors have deleted those fine details that are not relevant. 

5. Again I do not see the point about care givers reporting poor health and psychological instability etc………..in paragraph 4. May be authors should concentrate on older people themselves and what the evidence is, about their challenges.

We decided to leave the caregivers information because that offer some understanding on the current state of care needs of older adults. For instance, will caregivers avail themselves to care for older adults? Are caregivers going through circumstance that will be challenging for them to care for their ageing care recipients? 

6. In paragraph 5, the authors state ‘globally, there are a number of studies…..’ the introduction has to be rearranged. This global piece was brought in after issues in Ghana were earlier introduced. Typically, the global aspects will be introduced then narrowed down to Africa and Ghana. This will make reading of manuscript better.

Thanks for this comment. We have updated the entire introduction. 

7. Again authors bring in WHO-ICF framework closer to the end of introduction. This does not quite fit in here. This should be bought up earlier and discussed in a way that shows the reasons for its usage.

Authors have separated this under analytical framework as the aim was to help in the categorisation of the variables under all components. 

Methods

1. The authors state that the minimum sample size required in this study was 200. Readers are unable to tell how this was arrived at. Which sample calculation method was used? The sampling techniques used are unclear. In one aspect, authors state consecutive sampling using hospital register and later states random selection of patients based on days they visited the hospital. This part should be clarified in methods.

Thank you for your insight. Please we have addressed these comments. We have specified how the sample size was calculated. It now reads “The minimum sample size required in this study was 200 at a confidence level of 95% using Epi Info software (version 7.2.3). However, we increased this to 400 to compensate for any loss of participants.”

Moreover, we used two sampling techniques in this study. First, we used random sampling to select the days for data collection and after that consecutive sampling technique was used to select the participants based on the days they visited the hospital. We have clarified this sentence in the methods. 

2. Data collection. It will help if authors can add questionnaire as an appendix.

Thanks much. We have added the specific questions we used for this paper. 

3. This work has a 52% similarity with other works (checked with ithenticate). Authors should kindly modify to reduce the percentage of similarity with other published works.

Thanks for this comment. Authors have taken drastic measure to reduce the similarity percentage. Thanks 

4. Ethical considerations: I am unable to find statement on ethics in this work.

I have included a separate section on ethical consideration. 

Reviewer #2: 

i. In the abstract, kindly indicate the research gap calling for the study.

Thanks for this comment. We have provided the research gap in the abstracts. It now reads as “Given the longevity noticed among older people in Ghana, and the potential occurrence of functional disability in later years of lives, it has become essential to understand their care needs.”

ii. Under the data analysis section, the authors indicate that “ any variable with a p-value of 0.2 in the bi-variate association was considered for inclusion in the multivariate analysis” Is there any basis or theoretical justification for this? This should be discussed in the manuscript.

In public health discipline selecting a p-value of 0.2 is a common practice (Awuviry-Newton, Tavener, Wales, & Byles, 2020).

iii. In the methods, I did not see how the authors did measure validity and reliability of the data collection tool/ tool. 

International consistency for the group variable (functional disability) was assessed and it was found reliable and valid. Please see under “Activity limitations”.

iv. In the discussion, this is how the authors should discuss the results on the prevalence of care needs.

a. The authors should tell readers whether the prevalence of care needs reported in this study is higher than what has been reported elsewhere and then assign reasons for the differences. The authors should tell readers whether the prevalence of care needs reported in this study is the same as what has been reported elsewhere.

Thanks for this suggestion. Authors have addressed these comments in the discussion. We could not get results on prevalence of care needs per say. However, we had some evidence to extend the discussion.

b. The authors should tell readers whether the prevalence of care needs reported in this study is lower than what has been reported elsewhere and then assign reasons for the differences.

We have addressed this comment. 

v. A paragraph of the discussion should be devoted for the specific contribution of the study to literature.

This suggestion has been addressed. 

vi. The authors should also discuss the strengths of the study.

Thank you. We have addressed this comment. 

vii. The authors should further discuss how the limitations of the study did not affect the findings of the study.

Thanks. We have addressed this comment. 

viii. In the conclusion section, the authors did indicate that the prevalence of care needs is high among hospitalized older people. Is there any theoretical or conceptual basis for determining the prevalence at which care needs are high?

Thank you for this comment. This was the overall conclusion, which is thoroughly discussed in the discussion session. Thanks

---

## [Decision Letter · Decision Letter 1]

17 Dec 2021

PONE-D-21-10419R1Prevalence and determinants of care needs among older people in GhanaPLOS ONE

Dear Dr. Awuviry-Newton,

Thank you for submitting your manuscript to PLOS ONE. After careful consideration, we feel that it has merit but does not fully meet PLOS ONE’s publication criteria as it currently stands. Therefore, we invite you to submit a revised version of the manuscript that addresses the points raised during the review process.

We look forward to receiving your revised manuscript.

Kind regards,

Mohammad Bellal Hossain

Academic Editor

PLOS ONE

Reviewers' comments:

Reviewer's Responses to Questions

**Comments to the Author**

1. If the authors have adequately addressed your comments raised in a previous round of review and you feel that this manuscript is now acceptable for publication, you may indicate that here to bypass the “Comments to the Author” section, enter your conflict of interest statement in the “Confidential to Editor” section, and submit your "Accept" recommendation.

Reviewer #1: (No Response)

Reviewer #2: All comments have been addressed

2. Is the manuscript technically sound, and do the data support the conclusions?

Reviewer #1: Partly

Reviewer #2: Yes

3. Has the statistical analysis been performed appropriately and rigorously? 

Reviewer #1: Yes

Reviewer #2: Yes

4. Have the authors made all data underlying the findings in their manuscript fully available?

Reviewer #1: Yes

Reviewer #2: Yes

5. Is the manuscript presented in an intelligible fashion and written in standard English?

Reviewer #1: Yes

Reviewer #2: Yes

6. Review Comments to the Author

Reviewer #1: The manuscript has been significantly improved and reads better. However, there are some concerns.

1. Some editing is needed for example where authors mention ‘….a primary care giver’ this should be primary care givers.

2. ‘Often, these caregivers offer care and support with higher cost and burden, casting doubt on the continuity of their care’ . This sentence should be clarified as I do not see how cost of care cast doubt on care.

3. ‘… will assist in the development of an intervention to assist older people and relieve..’ this seems to be repetition as authors already stated this earlier …’ to provide data to assist policy and program developers to provide the appropriate health and social interventions’

4. ‘However, none of these studies were conducted in an African country, particularly in Ghana’. This statement may be misleading as there are number studies on care needs of people in Africa and Ghana. Authors could take a look at some these studies and show how their method is an improvement on previous methods.

5. I am also concerned with how much space is given to the framework in this manuscript. Manuscripts for publication in this case can discuss findings without dwelling so much on framework.

6. ‘However, we increased this to 400 to compensate for any loss of participants’ I do not see how increasing participants to 400 justify potential lost of participants. What did authors do at the recruitment stage that resulted in more sample. Probably that will explain this part better

Reviewer #2: Thank you for the opportunity to re-review the Manuscript Number PONE-D-21-10419R1. This is to inform you that the authors have successfully addressed all my previous comments and I have no further comments to raise. I am of the view that the manuscript is now ready for publication.

7. PLOS authors have the option to publish the peer review history of their article (what does this mean?). If published, this will include your full peer review and any attached files.

Reviewer #1: No

Reviewer #2: No

---

## [Author Response · Author response to Decision Letter 1]

21 Jan 2022

Response to reviewer’s comment

Dear editor,

Author’s of this manuscript offers their utmost gratitude for taking the time to given in-depth comments to improve the paper. We acknowledge that we have addressed all reviewer’s comments thoroughly and this have substantially improved the manuscript. 

Reviewer #1: 

The manuscript has been significantly improved and reads better. However, there are some concerns.

1. Some editing is needed for example where authors mention ‘….a primary care giver’ this should be primary care givers.

Response: Thanks for this, we have subjected the manuscript to English language editing. 

2. ‘Often, these caregivers offer care and support with higher cost and burden, casting doubt on the continuity of their care’. This sentence should be clarified as I do not see how cost of care cast doubt on care.

Response: This sentence has been retained because, however, we made few change in this. Please note that the opportunity cost of caring for older adults is enormous. 

3. ‘… will assist in the development of an intervention to assist older people and relieve.’ This seems to be repetition as authors already stated this earlier …’ to provide data to assist policy and program developers to provide the appropriate health and social interventions ‘a

Response: thanks to the reviewer. We have modified this expression to read better. 

4. ‘However, none of these studies were conducted in an African country, particularly in Ghana’. This statement may be misleading as there are number studies on care needs of people in Africa and Ghana. Authors could take a look at some these studies and show how their method is an improvement on previous methods.

Response: 

Thanks much we have addressed these comments.

5. I am also concerned with how much space is given to the framework in this manuscript. Manuscripts for publication in this case can discuss findings without dwelling so much on framework.

Response: thanks for your suggestion, however, authors intended to discuss the framework to inform the selection of the variable. We therefore think that it is ok to be in the manuscript.

6. ‘However, we increased this to 400 to compensate for any loss of participants’ I do not see how increasing participants to 400 justify potential lost of participants. What did authors do at the recruitment stage that resulted in more sample. Probably that will explain this part better

Response: this sentence was constructed in error. We have now updated the sentence and it as “However, we increased this to 400 to compensate for any probable loss of response for questions included in this study.”

Reviewer #2: Thank you for the opportunity to re-review the Manuscript Number PONE-D-21-10419R1. This is to inform you that the authors have successfully addressed all my previous comments and I have no further comments to raise. I am of the view that the manuscript is now ready for publication.

Response: 

Thanks much for your in-depth review. We acknowledge that your comments have substantially improved the manuscript.

---

## [Editor Report · Decision Letter 2]

2 Feb 2022

Prevalence and determinants of care needs among older people in Ghana

PONE-D-21-10419R2

Dear Dr. Awuviry-Newton,

We’re pleased to inform you that your manuscript has been judged scientifically suitable for publication and will be formally accepted for publication once it meets all outstanding technical requirements.

Kind regards,

Mohammad Bellal Hossain

Academic Editor

PLOS ONE
---

## [Editor Report · Acceptance letter]

7 Feb 2022

PONE-D-21-10419R2 

Prevalence and determinants of care needs among older people in Ghana 

Dear Dr. Awuviry-Newton:

I'm pleased to inform you that your manuscript has been deemed suitable for publication in PLOS ONE. Congratulations! Your manuscript is now with our production department. 

Kind regards, 

on behalf of

Dr. Mohammad Bellal Hossain 

Academic Editor

PLOS ONE